# Potential Impacts of *Prunus domestica* Based Natural Gum on Physicochemical Properties of Polyaniline for Corrosion Inhibition of Mild and Stainless Steel

**DOI:** 10.3390/polym14153116

**Published:** 2022-07-30

**Authors:** Muhammad Kamran, Anwar ul Haq Ali Shah, Gul Rahman, Salma Bilal

**Affiliations:** 1Institute of Chemical Sciences, University of Peshawar, Peshawar 25120, Khyber Pakhtunkhwa, Pakistan; kamichemist@yahoo.com (M.K.); gul_rahman47@uop.edu.pk (G.R.); 2National Centre of Excellence in Physical Chemistry, University of Peshawar, Peshawar 25120, Khyber Pakhtunkhwa, Pakistan

**Keywords:** *Prunus domestica* gum (PDG), PANI, composites, corrosion protection

## Abstract

The lack of an eco-friendly approach towards application of polyaniline as a coating material has been one of the most challenging tasks. Herein, the synthesis of green *Prunus domestica* gum grafted polyaniline (PDG-g-PANI) composite is reported by a cost-effective emulsion polymerization for application as an efficient anti-corrosion material for mild steel (MS) and stainless steel (SS) in a strong corroding environment. The composite formation was confirmed by Ultraviolet Visible (UV-Visible) and Fourier Transformed Infrared (FTIR) spectroscopies. X-ray diffraction data revealed the amorphous nature of the PDG-g-PANI. Scanning Electron Microscopic (SEM) images showed a bi-layered structure having a parent porous layer of PANI coated with afibrous layer of PDG. The solubility test confirmed the dissolution of PDG-g-PANI in common organic solvents such as acetone, ethanol, propanol, butanol, chloroform, N-Methyl-2-pyrrolidone, dimethyl sulfoxide, and the mixture of propanol and chloroform. The polarization curve, open circuit potential, electrochemical impendence spectroscopy (EIS), and gravimetric analysis were applied to investigate the corrosion protection behavior of the composite on MS and SS in 3.5% NaCl and 1 M H_2_SO_4_ solution. The PDG-g-PANI-coated MS exhibited 96% corrosion inhibition efficiency as compared to 86% and 43% for pristine PANI and PDG in 3.5% NaCl solution while PDG-g-PANI-coated SS showed 98% corrosion inhibition efficiency. Moreover, 99% and 96.6% corrosion protection was observed for PDG-g-PANI-coated MS and SS in 1 M H_2_SO_4_ solution. Gravimetric studies revealed that PDG-g-PANI coating can protect MS up to 93% for 14 days in salt solution while 97% corrosion inhibition efficiency was retained for 2 months in open air.

## 1. Introduction

Degradation of steel structures due to corrosion is one of the most crucial issues for many industries because it leads to rigorous reduction of investment and production loss [1]. “NACE international global study on corrosion costs and preventative strategies” in 2013 anticipated the global cost of corrosion to approximately 3.4% of worldwide GDP. To resolve this alarming issue, different strategies including corrosion inhibition, design of metal substrates, and organic coatings have been employed [2]. Different types of coating materials including polymers [3], natural materials [4], metal oxides, and their composites [5], have been extensively studied as steel corrosion inhibitors [6,7,8]. Among polymeric materials, polyaniline (PANI) has been widely applied for anticorrosion coatings for the last three decades, since DeBerry electrochemically deposited PANI on stainless steel which exhibited tremendous inhibition against corrosion by anodic protection, and by making an oxide layer on the metal surface, hence reducing the penetration of corrosive medium [9]. After that, different forms of PANI and their composites were applied as corrosion protection of metals [10]. Until now, two forms of PANI, a non-conducting emeraldine base (EB, half-oxidized form) and the conducting emeraldine salt (ES, half oxidized and protonated form) have been the most comprehensively investigated as anticorrosion coating materials due their tremendous redox reversible nature. Diverse results of corrosion protection and inhibition mechanisms have been reported, depending on the type of metal used, the method of PANI synthesis, the corrosion environments, and the coating methods [11]. However, some deficiencies related to PANI as the coating material still exist; for instance, its insolubility in common organic solvents owing to the rigidity of its conjugate chain, water adsorbing ability due to the presence of amino groups which enables the electrolyte to pass through the PANI coating to the metal surface. These deficiencies limit the applications of PANI as a coating material, most commonly in aqueous media [12]. To address these limitations, composites of PANI with other materials, especially synthetic polymers have been thoroughly investigated [13]. However, biosafety of the synthetic polymer has been a matter of concern [14].

Naturally occurring inhibitors have emerged as strong eco-friendly anti-corrosion agents for different metals and alloys [15,16]. One class of such materials are gums which are perceived as promising corrosion inhibitors due to their cost competitiveness, durability, and high mechanical strength [17]. Palumbo et al. applied gum arabic as a corrosion protection material for N80 steel in 0.5 M KCl and 68.78% inhibition efficiency was observed [18]. Eddy et al. stated that gums have good corrosion protection ability on metal surfaces due to the formation of complexes with metal ions through their electron-rich functional groups, which blanket and shield the metal from corrosive media due to the large surface area. The presence of -OH and -COOH functional groups, in most gums, increases the adsorption by contributing electrons or charge with the metal. They observed 76% inhibition efficiency while coating *Anogessus leocarpus* gum on mild steel in 0.1 M HCl solution [19]. Mobin et al. also worked on the application of natural material based on Xanthum gum for coating mild steel against corrosion in 1 M HCl solution and observed inhibition efficiency of 74.24% [20]. Aslam et al. presented a new green material based on multi-phytoconstituents from *Eucalyptus* bark extract for corrosion inhibition of mild steel in a saline environment. They observed an inhibition efficiency of 98.2% in 5% HCl solution [21]. Anh et al. prevented corrosion of steel items by coating a leaf extract of *Ficus racemosa* with inhibition efficiency of 93% in an acidic medium [22]. On the other hand, a composite of green plant material with PANI for corrosion protection of metals has arisen as a new interesting topic recently. Rashid et al. coated a polyaniline/palm oil blend for the corrosion protection of mild steel in a saline environment [23]. Introduction of xanthum gum to polyaniline showed better inhibition efficiency of 91% than both pristine gums in 1 M HCl [24].

In this work, *Prunus domestica* plant gum grafted PANI was studied for application as a corrosion protection material for steel surfaces. *Prunus domestica* gum (PDG) was obtained from a plum tree of the Roscaceae family. The PDG contains 73% water-soluble and 23% non-soluble fractions. PDG is mainly composed of galactose, arabinose, and minor fractions of xylose, and mannose with a molecular weight of 4.9 × 10^3^ kDa [25]. This project might be considered as a green protocol in which a PDG-grafted PANI was synthesized using a cost competitive emulsion polymerization in order to formulate a good coating material for corrosion protection of MS and SS. The material is eco-friendly, and the method of preparation is cost effective. The material meets the requirements for an efficient coating material to prevent enormous loss of steel substrate which has tremendous application in every field of life.

## 2. Experimental

### 2.1. Materials

Double distilled aniline (C_6_H_5_NH_2_) from Sigma Aldrich (St. Louis, MO, USA) was used as the monomer. *Prunus domestica* gum (PDG) was collected from a plum tree located in Village Shakarpura, Tehsil Shah Alam, District Peshawar, Pakistan. After collection, grains of gum were washed several times with distilled water followed by drying at room temperature for 2 weeks. After complete drying, it was crushed into fine powder and stored for further use. Commercial grade diesel was purchased from Pakistan state oil (PSO), with a boiling range of 160–366 °C, 0.828 specific gravity, and 3.11 cst viscosity at 20 °C, and used without any further treatment. Dodecylbenzenesulfonic acid (DBSA) (C_12_H_25_C_6_H_4_SO_3_H) (Acros organic, Morris, NJ, USA), benzoyl peroxide (C_14_H_10_O_4_) (Merck, Kenilworth, NJ, USA), chloroform (CHCl_3_), n-hexane, methanol, ethanol, propanol, 2-butanol, acetone, NMP, and DMSO were purchased from Sigma Aldrich (St. Louis, MO, USA), and doubled distilled water was used throughout the study.

### 2.2. Synthesis of PDG-Grafted Polyaniline (PDG-g-PANI)

In a typical experiment, 30 mL distilled water was taken in a conical flask, followed by addition of 0.1 g PDG powder under constant stirring. After 30 min of continuous stirring, 1 mL DBSA was added followed by the addition of 30 mL diesel. To this mixture, 0.4 mL aniline monomer was added and left stirring for 15 min till the formation of a white milky emulsion. Then, 1 g BPO oxidant was added to the reaction mixture and the flask opening was sealed and left stirring for 24 h at room temperature. After 24 h of stirring at room temperature, a blackish green product was formed in the form of precipitated grain or ball or in some cases the product was deposited on the walls of a conical flask. After the completion of the reaction, the liquid mixture was separated and put in a separating funnel while the precipitated product was washed with distilled water several times in order to remove water-soluble impurities followed by addition of acetone in order to disperse the product. The solution was poured into a porcelain dish and was dried at room temperature. Finally, the material was washed with n-hexane to remove non-polar impurities and then kept at 40 °C for 24 h.

### 2.3. Optimization

The effect of different reactants such as PDG, aniline, oxidant, and surfactant on the final product were optimized through corrosion studies by carrying out potentiodynamic polarization measurements. The best products with good corrosion inhibition properties were selected for further studies. The results of optimization are presented in the Appendix A.

### 2.4. Characterization

The solubility of the composites was determined in various common organic solvents including methanol, ethanol, propanol, butanol, acetone, chloroform, NMP, *n*-hexane, and DMSO. The PDG-g-PANI was characterized by UV-Vis spectroscopy with the help of a UV-Vis 1800 Shimadzu double beam spectrophotometer using chloroform as the solvent in 10 mm quartz cuvettes. For confirmation of structural information, FTIR spectra were recorded with an IR Affinity-S1 (Shimadzu, Kyoto, Japan) spectrophotometer, scanning in the wave number range of 400–4000 cm^−1^, with 2 cm^−1^ resolution. XRD patterns of samples were recorded through Cu Kα radiations (λ = 1.5405 A°) on a Rigaku (Kyoto, Japan) X-ray diffractometer with a progressive scan rate of 0.05°/s. Surface morphologies were investigated with a scanning electron microscope (JSM-5910, JEOL).

### 2.5. Corrosion Study

The corrosion protection performance of the composites was analyzed in an intense corrosive solution of 3.5% NaCl, 1 M H_2_SO_4_, and in open atmosphere. MS and SS coupons were purchased from local market and were carved into disc electrodes with a thin handle of surface diameter 10 mm. The elemental analysis of the coupons made from mild steel showed that it contained 1.28% C, 0.73% Mn, and 97.99% Fe by weight. However, the chemical composition of electrode made from stainless steel is different from MS containing 2.82% C, 0.36% Si, 8.15% Cr, 12.04% Mn, 1.59% Ni, and 75.03% Fe by mass.

Corrosion experiments were carried out in a single compartment cell containing three electrodes, using Reference 3000 potentiostat/galvanostat (Gamry, Warminster, PA, USA). MS and SS discs were used as working electrodes. A stainless steel plate and saturated calomel electrode (SCE) were used as counter and reference electrodes, respectively. Before coating, steel electrodes were cleaned with emery paper of 120–800 grit sizes followed by polishing with a polishing cloth containing alumina fine powder (0.05 micron) on its surface. Then, the electrode was washed with a mixture of ethanol and acetone. The composite dissolved in chloroform and was drop-coated on the steel electrode to form a uniform layer. The potentiodynamic polarization measurements were performed by scanning the potential in the range of ±400 mV (cathodic/anodic plots). Various parameters such as corrosion current density (*i_corr_*), corrosion potential (E_corr_), and corrosion rate (CR mm/year) of the steels were evaluated by Tafel extrapolation using Gamry Echem Analyst software. Inhibition efficiency of coatings was determined by using the following Equation (1) [1].
(1)$$\mathrm{Inhibition}\;\mathrm{Efficiency}\;(I.E)\;\%=1-\frac{i_{corr}}{i_{corr}^{0}}\;\times \;100$$
where “*i_corr_*” is the corrosion current density of coated steel electrodes while $i_{corr}^{0}$ is the corrosion current density of uncoated steel electrodes. EIS measurements were also carried out to analyze the resistive and capacitive properties of the samples. Open-circuit potential (OCP) was analyzed over a wide range of frequencies (f) from 0.05 Hz to 100 kHz to collect EIS data and presented in the form of Nyquist and Bode plots. The corrosion study was also performed by weight loss method. For this study, two electrodes were polished with emery paper and then with alumina powder. After that, surfaces of both electrodes were degreased with ethanol and acetone followed by distilled water and dried in an oven and weighed. After that, one electrode was coated with composite, following that both coated and bare electrodes were dipped in salt solution and kept for a specific amount of time. The same process was also repeated both for coated and uncoated steel electrodes in open air as well, for the specific amount of time. Corrosion rate was calculated by measuring weight loss using the following relationship [26].
(2)$$\mathrm{Corrosion}\;\mathrm{rate}=\frac{W}{DAt}$$
where *W* is weight loss in g, *D* is density in g/cm^3^ (=7.8 g/cm^3^), *A* is area in cm^2^, and *t* is time of exposure in seconds (s). In order to convert the corrosion rate to m/year, 3.15 × 10^8^ was used as the multiplication factor.

While inhibition efficiency was calculated by using the following equation.
(3)$$\mathrm{Inhibition}\;\mathrm{efficiency}\;(\%)=[\frac{CR^{o}-CR}{CR^{o}}\times 100]$$
where *CR^o^* is corrosion rate of the uncoated steel electrode and *CR* is the corrosion rate of the coated steel electrode [27].

## 3. Results and Discussion

### 3.1. Solubility

The solubility of synthesized PDG-g-PANI was checked in different common organic solvents including polar methanol, ethanol, propanol, butanol, N-methyl-2-pyrrolidone (NMP), dimethylsulfoxide (DMSO), and nonpolar acetone and chloroform and the mixture of these solvents as shown in Figure 1. The solubility of PDG-g-PANI can be affected by the interactions among PANI chains, PDG units, the associated counter ions of dopant, and the solvent. Results showed that PDG-g-PANI is readily soluble in all above-mentioned polar and nonpolar organic solvents. This can be explained by the fact that there are various polar and non-polar moieties in the structure of PDG-g-PANI which make it soluble in both organic and aqueous solvents. For instance, the material is doped with DBSA containing a long non-polar alkyl chain and a polar –HSO_3_ group. Similarly, the nitrogen atom of aniline units and hexagonal rings of PDG containing hydroxyl groups can help in hydrogen bonding. The presence of these groups in PDG-g-PANI imparts solubility in polar as well as non-polar organic solvents [28].

### 3.2. UV-Vis Spectroscopy

UV-Vis spectra of PDG, PANI, and PDG-g-PANI are depicted in Figure 2. The UV-Vis spectrum of PDG shows a continuous absorption throughout the visible region and gives a clear peak at 272 nm. This peak is assigned to the arabinose and galactose components present in the PDG [29,30]. The UV-Vis spectrum of PANI shows peaks at 276 and 402 nm, and a broad hump at 600 nm. The peak at 276 nm is due to π-π* transition of benzoid rings of PANI. The peak at 402 nm is assigned to n-π* transition of the quinoid ring while the broad peak at the 600 nm region is assigned to π-polaron transition also called exiton [31]. The UV-Vis spectra of PDG-g-PANI exhibit four peaks having three characteristics peaks of pristine PANI and a respective peak of PDG. A broad hump in the 600 nm region can be attributed to the transition of exiton of the polymer chain, while the peak at 400 nm region is assigned to n-π* transition of quinoid rings while a couple of peaks at 300 and 260 nm regions are attributed to π-π* of benzoid rings and arabinogalactose components of PDG. All these peaks indicate the formation of PDG-g-PANI [32].

### 3.3. FTIR Spectroscopy

FTIR spectra of pure PDG, PANI, and PDG-g-PANI are shown in Figure 3. The spectrum of PDG showed a broad band at 3291 cm^−1^ which is due to O–H stretching vibrations. The peak at 2915 cm^−1^ is due to stretching vibration of the aliphatic C–H bond of arabinose and galactose units. The band at the 1791 cm^−1^ band is attributed to C=O stretching of –COOH, while the conjugated C=C bond stretching vibration gives a peak at 1605 cm^−1^. The peaks at 1413 and 1023 cm^−1^ are assigned to C–O stretching vibrations [33,34]. From the FTIR spectrum, it can be concluded that PDG contains mainly polysaccharide compounds, having –OH, −COOH, ether, and hemiacetal groups. The FTIR spectrum of PANI shows peaks at 3241 cm^−1^ which is due to symmetric and asymmetric stretching vibrations of NH_2_ and NH [35]. The bands at 2930 and 2851 cm^−1^ are assigned to C–H stretching vibrations of the aniline ring while the characteristic peaks of quinoid ring C–N stretching and the benzenoid ring appeared at 1583 and 1489 cm^−1^, respectively. The bands that appeared at 1303 and 1170 cm^−1^ can be assigned to C–N stretching and bending vibration of C–H of the aniline ring of the polymer chain. The peaks at 1008 and 573 cm^−1^ simultaneously confirmed the doping of DBSA into the polymer chain, the former band appears due to –SO_3_H while the later peak appears due to the movement of –SO_3_ [36]. In the spectrum of PDG-g-PANI, the peak at the 3200 cm^−1^ region disappeared which may be assigned to hydrogen bonding of arbinogalactan of PDG with nitrogen of the aniline ring of the polymer chain. The bands at 2921 and 2851 cm^−1^ are assigned due to coupled stretching vibration of aliphatic C–H and C–H of the aniline rings of the polymer chain. The emergence of the typical peak of COOH due to stretching vibration of –C=O at 1716 cm^−1^ confirms the grafting of PDG into the polymer chain. The peak corresponding to C-N stretching of the quinoid ring slightly shifted to 1593 cm^−1^ is due to the interaction of nitrogen of quinoid and hydrogen of the carboxyl group of PDG components. Moreover, a shift was also seen in the stretching vibration of C–N of the benzenoid ring which appeared at 1498 cm^−1^. A coupled peak at 1000 cm^−1^ is due to stretching vibration of C–O of COOH and –SO_3_H of DBSA. The characteristic peak of –SO_3_ appeared at 573 cm^−1^.

### 3.4. X-ray Diffraction (PXRD) Analysis

The PXRD pattern of the PDG is shown in Figure 4. A broad peak appeared at the 2 Theta range of 17–19° and no sharp peak can be seen in the diffraction pattern which indicates the amorphous nature of PDG. XRD patterns of PDG-g-PANI samples (Figure 5) exhibited a broad peak in the 2θ region of 18–19° [37]. This peak can be ascribed to periodically perpendicular and parallel PANI chains in the composite. The absence of sharp peaks indicates the amorphous nature of the composites as well [29,38].

### 3.5. Scanning Electron Microscopy (SEM)

Scanning electron microscopy is a suitable approach which ascertains the size and surface texture of the polymer matrix. The SEM micrographs of pristine PANI, PDG, and PDG-g-PANI are shown in Figure 6. The micrographs of PANI show a smooth texture containing pores of different sizes. On the other hand, PDG shows an irregular shape having furrows and grooves on the surface. The SEM images of all optimized samples show almost similar surface texture with negligible difference in surface morphology. These micrographs show a bi-layered structure with a parent porous layer coated with a fibrous layer. The parent layer is that of polyaniline on which the fibrous layer of PDG is deposited. These morphologies clearly indicate the grafting of PDG into the polyaniline chain.

### 3.6. Corrosion Study

#### 3.6.1. Open Circuit Potential

The corrosion protection tendency of coating materials for mild steel (MS) was analyzed by open circuit potential “OCP” analysis. The variation in OCP with time for bare and coated MS in 3.5 wt% NaCl solution is shown in Figure 7. The initial OCP values for bare and PDG-g-PANI-coated MS electrodes were found to be −0.615 and −0.491 V, respectively. At the start of immersion of the PDG-g-PANI coating, the gradual decline in the potential is attributed to the permeation of electrolyte solution through the microdefects into the substrate/coating interface but the coating strongly protected MS from corrosion in a mild corrosive environment [39]. The OCP of the PDG-g-PANI coating showed stable potential as compared to the bare MS. The OCP analysis of the bare MS surface showed a turbulent model and took a comparatively longer time to stabilize [40]. After 6000 s, the potential of PDG-g-PANI-coated MS was −0.601 V Vs SCE while that of bare MS was noted to be −0.641 V. A prominent positive shift in the OCP of PDG-g-PANI-coated MS indicated that the surface of MS is efficiently protected by PDG-g-PANI against corrosion [39]. Generally, the occurrence of a high value OCP is correlated with the presence of a layer with higher corrosion protection behavior. In the case of PDG-g-PANI coating, along with its physical barrier behavior, it provides efficient protection for the steel surface. On the basis of the above analysis, it can be inferred that the PDG-g-PANI coatings protect mild steel effectively against corrosion in a saline medium [3].

#### 3.6.2. Electrochemical Impedance Spectroscopy (EIS)

EIS is an expedient and quick method for evaluation of the corrosion inhibitive properties of coatings which does not disturb the double layer at the metal/solution interface giving reliable results [41]. The corrosion resistance ability of PDG-g-PANI was determined by measuring EIS in 1 M H_2_SO_4_ solution at room temperature. A small controlled scratch was made in the coating in order to assess corrosion medium on the metal surface. Bode plots of uncoated MS and PDG-g-PANI-coated MS are shown in Figure 8. As indicated, the total impedance of PDG-g-PANI-coated MS is higher than that of the uncoated sample. Bode modulus renders that on the coating composite on MS, impedance increases as compared to uncoated MS indicating reduction in corrosion rate of MS [41,42]. The increase in the total impedance value after coating MS with PDG-g-PANI indicates the minimization of charge transfer from metal to solution phase and from solution phase to metal [27].

The Nyquist plots of the uncoated and PDG-g-PANI-coated MS are shown in Figure 8. The radius of the semicircle is directly related to corrosion resistance. The PDG-g-PANI-coated electrode showed higher charge transfer resistance as compared to the uncoated sample, indicating that the surface of MS is well protected against corrosion under the chemical corrosive environment [43]. Randles model was applied on uncoated and coated Nyquist plots for MS and impedance parameters were calculated from the fitted model. The results are given in Table 1, while inhibition efficiency was calculated by using the following equation.


(4)
$$\mathrm{Inhibition}\;\mathrm{efficiency}\;(\%)=[\frac{Rct^{o}-Rct}{Rct^{o}}\times 100]$$


It can be seen from the table that charge transfer resistance (*Rct*) increased from 4.046 to 208.9 ohm for PDG-g-PANI-coated MS. However, the solution resistance (Rs) remains almost constant while constant phase element (CPE) increases after coating MS with PDG-g-PANI. Corrosion inhibition efficiency of PDG-g-PANI on MS was 98.06 which is in close agreement with other techniques applied.

#### 3.6.3. Potentiodynamic Polarization Curve (Tafel Plot)

The potentiodynamic curves of uncoated MS, pristine PDG, PANI, and PDG-g-PANI-coated MS were measures in 3.5% NaCl solution, as shown in Figure 9. The essential corrosion kinetic parameters including corrosion potential (*E_corr_*), corrosion current density (*i_corr_*), anodic (βa) and cathodic (βc) slopes were calculated from the Tafel fit and presented in Table 2. The corrosion potential (*E_corr_*) of uncoated MS was recorded at −866 mV, while the corrosion current density was 21.90 μA/cm^2^. The corrosion rate of MS was 10 m/year as shown in Table 2. The coating of natural PDG on MS lowered both E_corr_ and *i_corr_* comparatively and gave values of −697 mV and 12.50 μA/cm^2^, respectively, while the corrosion rate was decreased to 5.68 m/year showing a corrosion protection efficiency of 43.2%. However, the solubility of PDG in water limits its application as a corrosion protection material, making it unfit as a coating material in aqueous medium. On the other hand, PANI-coated MS *E_corr_* was −630 mV and *i_corr_* was 3.09 μA/cm^2^ obtained after Tafel extrapolation. The corrosion rate was decreased to 1.410 m/year as compared to 10 m/year of uncoated MS showing an inhibition efficiency of 85.89%.

However, the coating of PDG-g-PANI on MS lowered the *i_corr_* by the process of synergism, which enhanced the corrosion inhibition performance [44]. On coating the MS with PDG-g-PANI, a clear positive shift was observed in the corrosion potential and was recorded to be −538 mV while the corrosion current density was reduced to a high extent (recorded at 0.691 μA/cm^2^) and the corrosion rate was reduced to 0.315 m/year, with a corrosion protection efficiency of 96.84%. The protection mechanism can be described in two ways: (i) PANI chain in the composite has a physical interaction with iron atoms on the steel surface due to nitrogen present in its chain and thereby passivated the surface of steel by redox mechanism. In this process, iron (Fe) is readily oxidized to Fe^2+^ and then to Fe^3+^ subsequently. The electron released is accepted by the PANI chain in Emeraldine salt (ES) form and is reduced to Lecuemeraldine (LE) form. At the same time, the released electrons of Fe can also reduce oxygen in water to “OH^−”^. This OH^−^ ion will react with Fe^+3^ to form a strong passivating layer of Fe_2_O_3_ and water. Additionally, the dissolved oxygen again oxidized the LE to ES form, hence protecting the surface of iron from corroding by a cyclic process [28,31,45,46]. The whole process is summarized in the equations below.

PANI-A (ES) + ne^−^ → PANI (EB) + A^n−^(5)

Fe → Fe^+2^ + 2e^−^(6)

2Fe^+2^ +3H_2_O → Fe_2_O_3_ + 6H^+^ + 2e^−^(7)

O_2_ + 2H_2_O + 4e^−^ → 4OH^−^(8)

PANI (EB) + An^−^ → PANI-A (ES) + ne^−^(9)

(ii) On the other hand, PDG in the composite helps with adsorption to the steel surface due to the presence of a large number of higher charge density oxygen atoms, and hence reduces the corrosion of the surface [47]. Due to presence of many electron-rich functional groups, it adsorbs on the surface of carbon steel electrostatically or by making complexes through a coordinate type of linkage by sharing of a lone pair of electrons with partially filled metal orbitals. The large surface area of PDG blankets the surface of carbon steel and makes a shield against the corrosive agent present in the solution. The presence of PDG in the coating makes it thermally less conductive and mechanically strong [4]. Li et al. and Ferreira et al. stated that “if the shift in corrosion potential of coated MS is more than 85 mV with respect to the corrosion potential of uncoated, the inhibitor can be considered as an anodic or cathodic type”. In the present study the maximum displacement of E_corr_ was less than 85 mV, indicating that the PDG-g-PANI is a mixed-type inhibitor [48,49].

The behavior of pristine PDG, PANI, and PDG-g-PANI coating was also checked on SS and the results are shown in Figure 10 and Table 3. The E_corr_ and i_corr_ values of SS in 3.5% NaCl solution recorded were −960 mV and 11.90 μA/cm^2^, respectively. The corrosion rate of SS was 5.421 m/year. Coating of PDG on SS reduces corrosion to 3.97 m/year with corrosion efficiency of 26.9%. PANI coating shows better behavior and inhibits the SS against corrosion in a strong corrosive environment of salt. The PANI coating shifted the corrosion potential to the positive side and recorded a potential of −459 mV in comparison to −960 mV for uncoated SS. The corrosion current density decreased to 1.56 μA with 0.714 m/year rate of corrosion. The inhibition efficiency of PANI on SS recorded was 86.89%. Coating the surface of SS with PDG-g-PANI prominently shifted the corrosion potential in a positive direction with a value of −573 mV with a reduction in corrosion current density to 0.232 mA/cm^2^. The PDG-g-PANI coating resulted in an inhibition efficiency of 98.05%. The data reflect the amazing corrosion protection behavior of PDG-g-PANI on both MS and SS.

#### 3.6.4. Effect of Temperature

To assess the effect of temperature on corrosion protection behavior of PDG-g-PANI coated on MS, potentiodynamic polarization experiments were performed at 10, 20, 30, 40, and 50 °C. The results obtained are given in Figure 11, which showed that the corrosion rate increased with the rise in temperature. However, the PDG-g-PANI coating can be seen to maintain its corrosion protection effect at all temperatures. It was observed that at low temperature the inhibition efficiency was high which decreased with the rise in temperature. A decrease in the inhibition efficiency of the coating with the rise in temperature implies that the surface active groups of the PDG-g-PANI coating are physically adsorbed on the MS surface. Physiosorption leads to weak inhibitor–metal interaction which can reduce its effect at high temperatures [50]. On the other hand, higher solution temperature disrupts hydrogen adsorption which has blocked the cathodic area at lower temperatures, leading to increased corrosion rate. The apparent activation energy (*Ea*) was calculated from the Arrhenius plot for uncoated and coated mild steel in 3.5% NaCl solution. Activation energies for the corrosion process were calculated using the Arrhenius equation given below [41].


(10)
$$\log\;\mathrm{CR}=\frac{-Ea}{2.303RT}+\log A$$


The Arrhenius plot of log CR versus 1/T of uncoated MS and PDG-g-PANI-coated MS is shown in Figure 11. The slopes were estimated and activation energy was calculated using the expression Ea = (slope) × 2.303R and given in Table 4. The Ea value in the presence of PDG-g-PANI-coated MS is higher than that of uncoated MS in salt solution, which can be attributed to the decrease in surface area available for corrosion.

#### 3.6.5. Corrosion Studies in Sulphuric Acid Medium

Corrosion studies were also carried in strong corrosive acidic medium. Sulphuric acid solution with a concentration of 1 M was used. The Tafel plot of uncoated MS and PDG-g-PANI-coated MS is shown in Figure 12, while the corrosion kinetic parameters obtained from the Tafel fit are given in Table 5. The anodic and cathodic slopes for uncoated MS are 0.159 and 0.209 V/decade, respectively. The corrosion current density for uncoated MS recorded was 1480 μA with a corrosion potential of −508 mV. The acidic medium corrodes the MS surface with the rate of 675.3 m/year which is much higher than that of NaCl medium. After coating the composite on MS, both the anodic slope and cathodic slope decreased to 0.077 and 0.200 V/decade. The corrosion current density of coated MS declines to 4.680 μA as compared to 675.3 μA with a shifting of corrosion potential to a positive direction and gave a value of −376 mV. The rate of corrosion falls prominently to 2.139 with an inhibition efficiency of 99.68%. These results indicate good corrosion protection ability of PDG-g-PANI on MS in the strongly corrosive medium of sulphuric acid.

On the other hand, the behavior of uncoated and coated SS was tested in 1 M H_2_SO_4_ solution. The results are given in Table 6 and Figure 13. Uncoated SS corroded overwhelmingly in a concentrated solution of H_2_SO_4_ with the rate of 2157 m/year with 4720 μA current density and with a corrosion potential of −491 mV. On coating SS with PDG-g-PANI, the corrosion current decreased tremendously with a value of 111 μA and decreased the rate of corrosion phenomenally to 50.84 m/year. PDG-g-PANI showed 97.64% inhibition efficiency in a stainless steel strong corrosive environment.

### 3.7. Kinetic and Gravimetric Studies

The gravimetric analyses were carried out by calculating the loss in weight after immersing the uncoated and PDG-g-PANI-coated MS in 3.5% NaCl solution for 14 days. Corrosion rates for both uncoated and coated electrodes were calculated using weight-loss values from the experiments. The corresponding inhibition efficiencies were calculated by using these corrosion rate values for both uncoated and PDG-g-PANI coating as the inhibitor and the results are given in Table 7. The weight loss for uncoated MS observed was 0.07 g after 14 days of immersion in 3.5% NaCl solution while that of PDG-g-PANI-coated MS was 0.005 g. After calculation, the corrosion rate of uncoated MS was 2.33 m/year and PDG-g-PANI-coated MS was 0.16 m/year with 93.13% inhibition efficiency. The results were in close agreement with electrochemical data.

#### Open Air Study

The gravimetric or weight loss measurements were also performed in open air. Two electrodes were labeled and polished by emery paper of different sizes followed by alumina powder on pad. The electrodes were degreased with acetone and distilled water. After this, one electrode was coated with PDG-g-PANI and the other was kept uncoated. Both the electrodes were dipped in 3.5% NaCl solution for 30 min and then kept in open air. The electrodes were sprayed with salt solution after each three days and left in open atmosphere for 2 months. After 2 months, the uncoated mild steel electrode was rusted while the coated electrode remained almost the same after removing the material, as shown in Figure 14.

The corrosion rates for both coated and uncoated electrodes were determined using the following equation [26] and summarized in Table 8


(11)
$$\mathrm{Cr}=\frac{W}{DAt}$$


The corrosion rate of uncoated MS after 2 months of exposure to open atmosphere was 0.1 m/year while that of PDG-g-PANI-coated MS was 0.0031 m/year with an inhibition efficiency of 97%. This indicated that PDG-g-PANI coating inhibits steel efficiently in open atmosphere.

## 4. Conclusions

PDG-g-PANI was studied for the anticorrosion behavior on mild steel and stainless steel in 3.5% NaCl, 1 M H_2_SO_4_ solution, and open air using different electrochemical and gravimetric techniques. PDG-g-PANI was successfully synthesized using a cost effective and green emulsion polymerization method and characterized using UV−Visible/NIR, FTIR, XRD, and SEM techniques. The solubility study showed that composites are completely soluble in common organic solvents such as acetone, ethanol, propanol, butanol, chloroform, NMP, and DMSO and their mixtures. Potentiodynamic polarization studies showed that the composite exhibits a mixed-type effect both on mild and stainless steel dissolution in different corroding environments. The composites effectively suppressed the corrosion of mild steel and stainless steel in 3.5% NaCl, 1 M H_2_SO_4_, and in open air, and protect the surface of steel items tremendously with a corrosion inhibition efficiency of 99%.

## Figures and Tables

**Figure 1 polymers-14-03116-f001:**
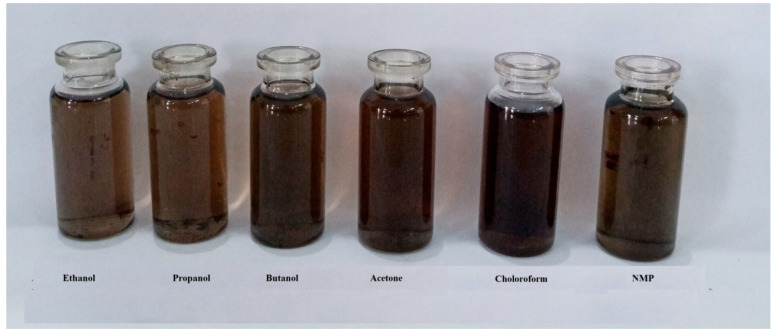
Solution of PDG-g-PANI in different organic solvents as indicated.

**Figure 2 polymers-14-03116-f002:**
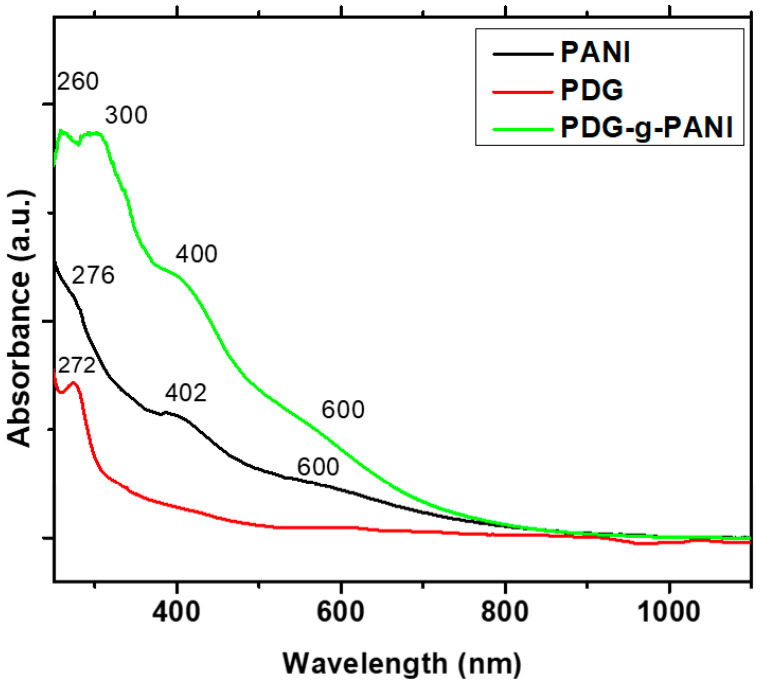
UV-Vis spectra of PDG, PANI, and PDG-g-PANI.

**Figure 3 polymers-14-03116-f003:**
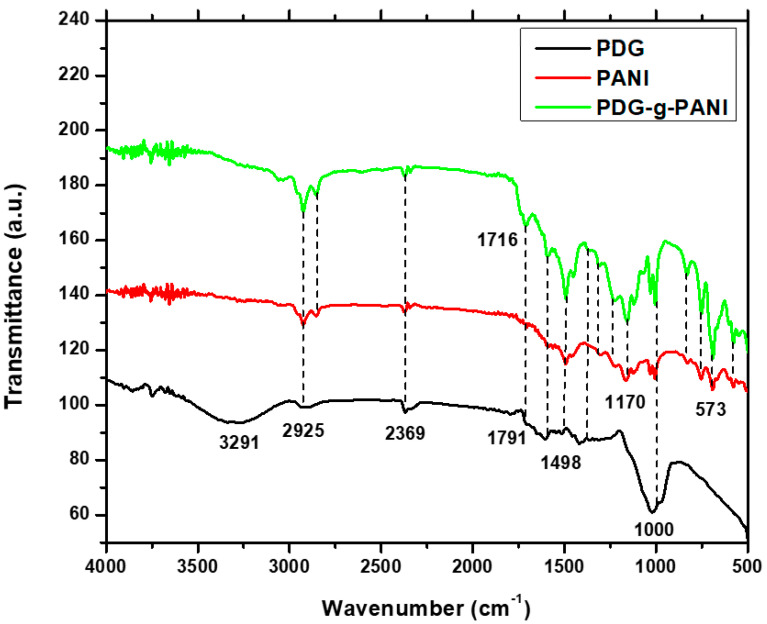
FTIR spectra of pristine PANI, PDG, and PDG-g-PANI.

**Figure 4 polymers-14-03116-f004:**
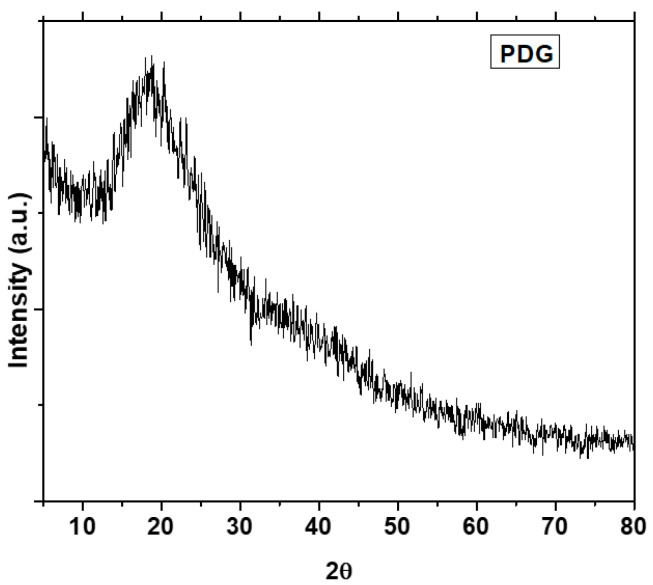
PXRD pattern of *Prunus domestica* gum powder.

**Figure 5 polymers-14-03116-f005:**
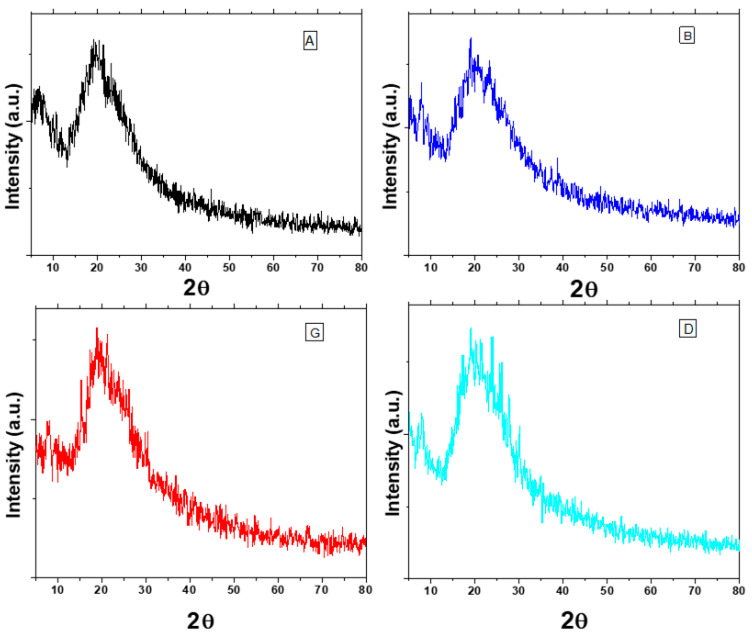
PXRD patterns of different PDG-g-PANI optimized products. (A) optimized with respect to aniline, (B) optimized with respect to BPO, (G) optimized with respect to gum and (D) optimized with respect to DBSA.

**Figure 6 polymers-14-03116-f006:**
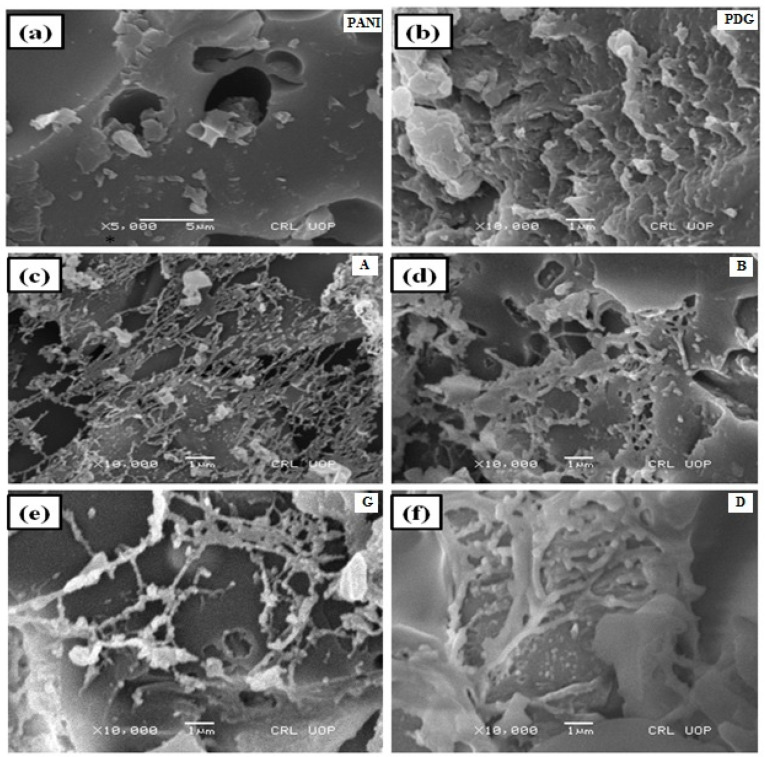
SEM images of (**a**) pristine PANI, (**b**) pristine PDG and PDG-g-PANI composites, (**c**) A = optimized with respect to aniline, (**d**) B = optimized with respect to BPO, (**e**) G = optimized with respect to gum, (**f**) D = optimized with respect to DBSA.

**Figure 7 polymers-14-03116-f007:**
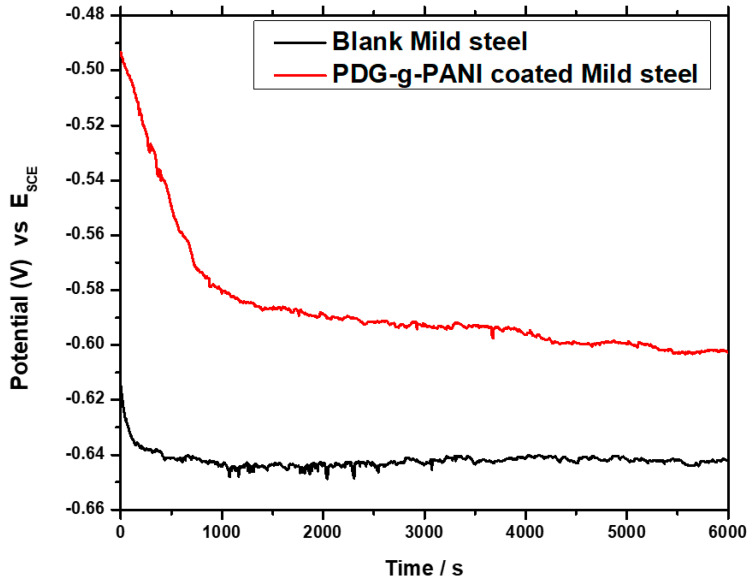
Open circuit potential of uncoated and PDG-g-PANI-coated MS.

**Figure 8 polymers-14-03116-f008:**
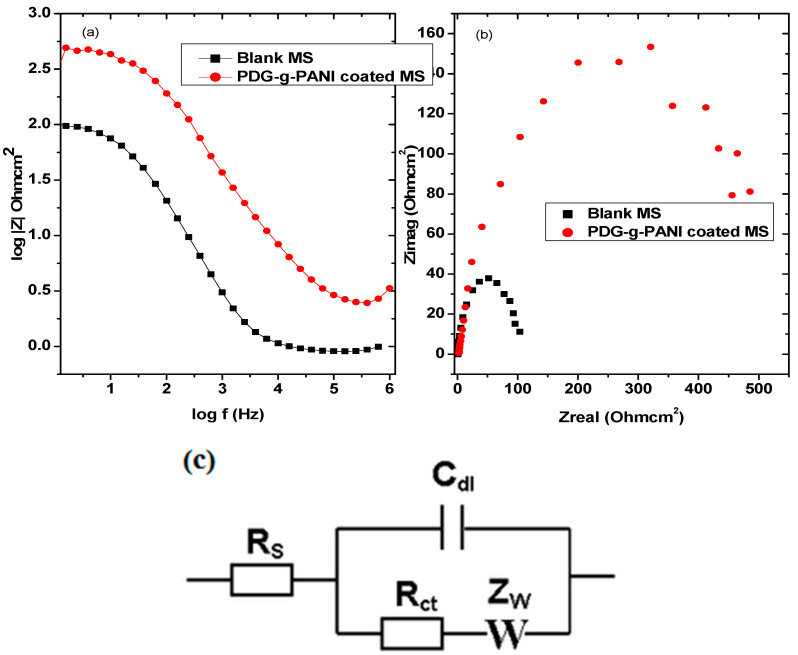
(**a**) Bode and (**b**) Nyquist plots of MS and PDG-g-PANI-coated MS, (**c**) Randles circuit.

**Figure 9 polymers-14-03116-f009:**
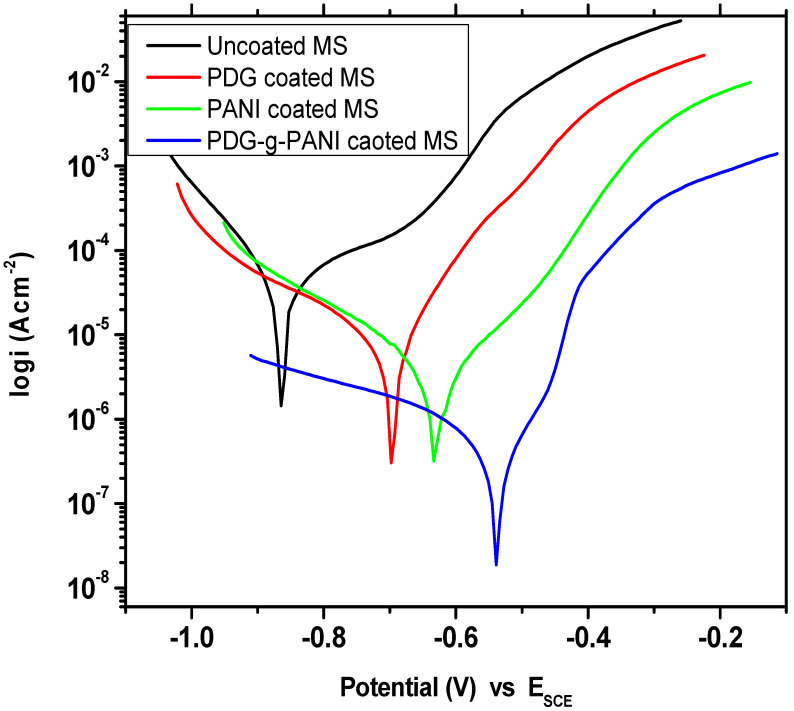
Tafel plot of uncoated MS, PDG-coated, PANI-coated, and PDG-g-PANI-coated MS.

**Figure 10 polymers-14-03116-f010:**
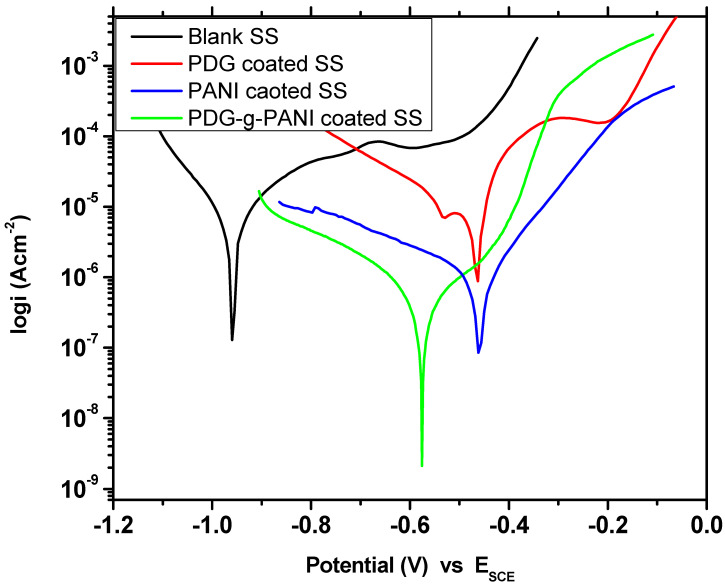
Tafel plot of uncoated SS, PANI-coated and PDG-g-PANI-coated SS.

**Figure 11 polymers-14-03116-f011:**
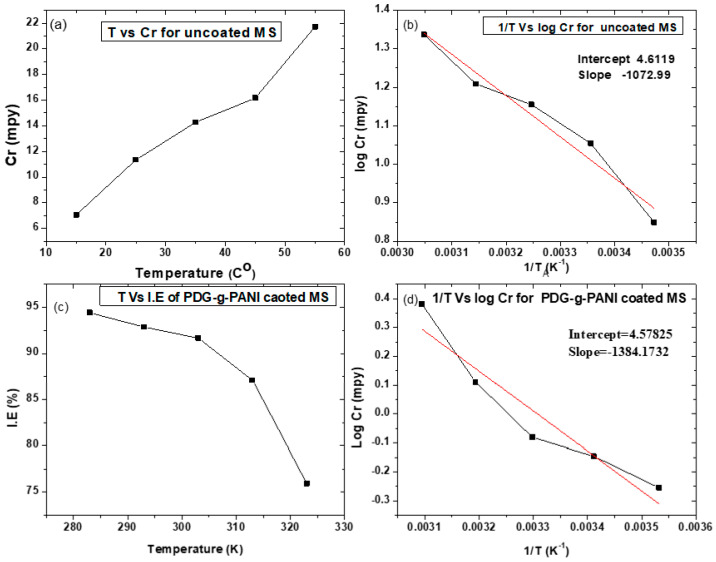
(**b**,**d**) Arrhenius plots of uncoated and PDG-g-PANI-coated MS corrosion in 3.5% NaCl solution, (**a**) temperature vs. corrosion rate graph for uncoated MS, (**c**) temperature vs. inhibition efficiency of PDG-g-PANI-coated MS.

**Figure 12 polymers-14-03116-f012:**
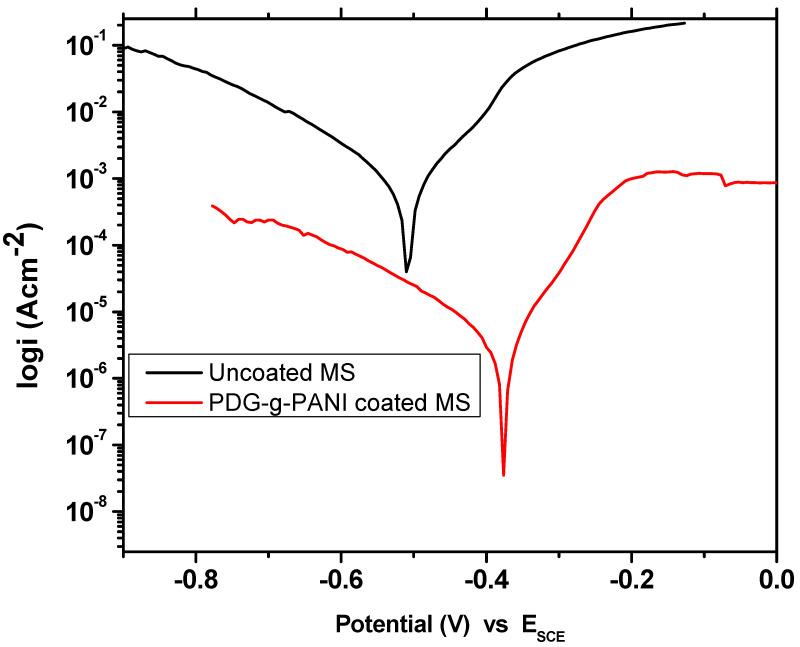
Tafel plot of uncoated MS and PDG-g-PANI-coated MS in 1 M H_2_SO_4_.

**Figure 13 polymers-14-03116-f013:**
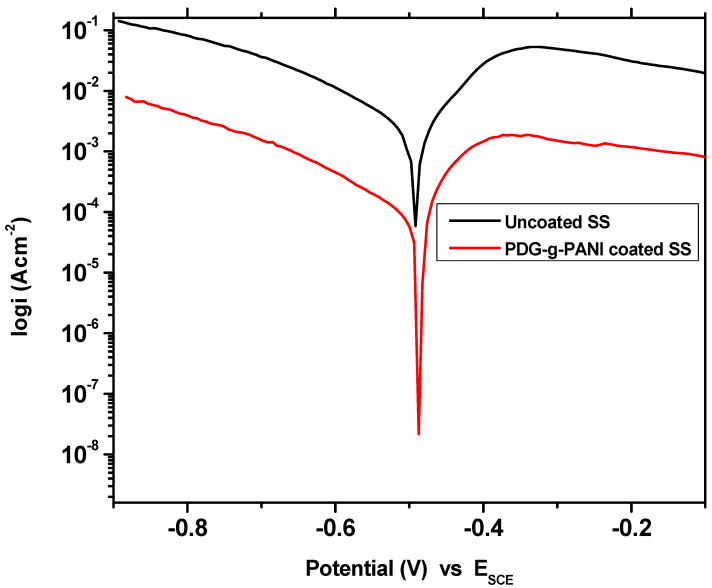
Tafel plot of uncoated SS and PDG-g-PANI-coated SS in 1 M H_2_SO_4_.

**Figure 14 polymers-14-03116-f014:**
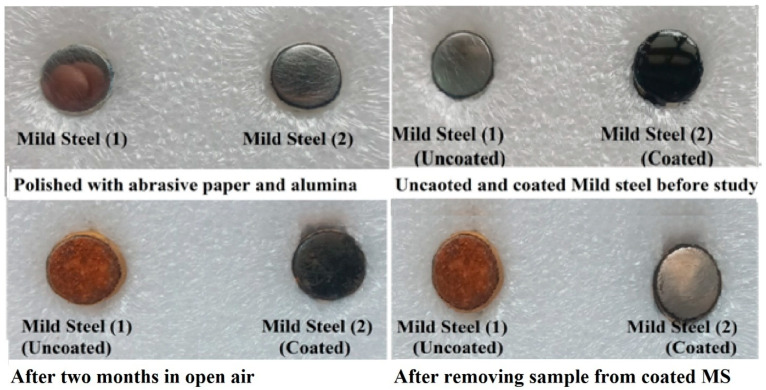
Pictures of uncoated and coated MS before and after the study in open air.

**Table 1 polymers-14-03116-t001:** Parameters obtained from Nyquist plot fitting of uncoated and coated MS.

Sample Name	Rct Ohm	Rs Ohm	CPE F	Inhibition Efficiency %
Uncoated MS	4.046	68.90	64.94 × 10^−6^	-
PDG-g-PANI-coated MS	208.9	78.70	3.696 × 10^−6^	98.06

**Table 2 polymers-14-03116-t002:** Comparison of essential corrosion kinetic parameters of uncoated MS, PDG-coated, PANI-coated, and PDG-g-PANI-coated MS in 3.5% NaCl solution.

Sample Name	Beta A V/Decade	Beta C V/Decade	I_corr_ μA/cm^2^	E_corr_ mV	Corrosion Rate (m/Year)	Inhibition Efficiency %
Uncoated MS	0.170	0.089	21.90	−866	10.00	-
PDG	0.135	0.297	12.50	−697	5.68	43.2
PANI	0.127	0.189	3.09	−630	1.410	85.89
PDG-g-PANI	0.119	0.407	0.691	−538	0.315	96.84

**Table 3 polymers-14-03116-t003:** Comparison of essential corrosion kinetic parameters of uncoated SS, PANI-coated, and PDG-g-PANI-coated SS in 3.5% NaCl solution.

Sample Name	Beta A V/Decade	Beta C V/Decade	I_corr_ μAcm^−2^	E_corr_ mV	Corrosion Rate (m/Year)	Inhibition Efficiency %
Uncoated SS	0.444	0.163	11.90	−960	5.421	-
PDG	0.184	0.265	8.70	−467	3.976	26.89
PANI	0.146	0.456	1.56	−459	0.714	86.89
PDG-g-PANI	0.109	0.180	0.232	−573	0.105	98.05

**Table 4 polymers-14-03116-t004:** Calculated activation energies of uncoated MS and PDG-g-PANI-coated MS.

Sample Name	$E_{act}$ (KJ)
MS	20.54
PDG-g-PANI-coated MS	26.50

**Table 5 polymers-14-03116-t005:** Comparison of essential corrosion kinetic parameters of uncoated MS and PDG-g-PANI-coated MS in 1 M H_2_SO_4_ solution.

Sample Name	Beta A V/Decade	Beta C V/Decade	I_corr_ μAcm^−2^	E_corr_ mV	Corrosion Rate (m/Year)	Inhibition Efficiency %
Uncoated MS	0.159	0.209	1480	−508	675.3	-
PDG-g-PANI-coated MS	0.077	0.200	4.680	−376	2.139	99.68

**Table 6 polymers-14-03116-t006:** Comparison of essential corrosion kinetic parameters of uncoated SS and PDG-g-PANI-coated SS in 1 M H_2_SO_4_.

Sample Name	Beta A V/Decade	Beta C V/Decade	I_corr_ μAcm^−2^	E_corr_ mV	Corrosion Rate (m/Year)	Inhibition Efficiency %
Uncoated SS	0.138	0.257	4720	−491	2157	-
PDG-g-PANI	0.092	0.203	111	−487	50.84	97.64

**Table 7 polymers-14-03116-t007:** Weight of uncoated and coated MS before and after immersion, with corresponding corrosion rate and inhibition efficiency of PDG-g-PANI.

Sample Name	Weight before Immersion (g)	Weight after Immersion (g)	Weight Loss (g)	Corrosion Rate (m/Year)	Inhibition Efficiency %
Uncoated MS	62.8049	62.7979	0.07	2.33	-
PDG-g-PANI-coated MS	61.2450	61.2445	0.005	0.16	93.13

**Table 8 polymers-14-03116-t008:** Weight of uncoated and coated MS before and after study in open air, with a corresponding corrosion rate and inhibition efficiency of PDG-g-PANI.

Sample Name	Weight before Exposure (g)	Weight after Exposure (g)	Weight Loss (g)	Corrosion Rate (m/Year)	Inhibition Efficiency %
Uncoated MS	48.8160	48.8028	0.0132	0.1	-
PDG-g-PANI	49.2429	49.2425	0.0004	0.0031	97

## Data Availability

Not applicable.

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
