# Peer review of "Potential Impacts of Prunus domestica Based Natural Gum on Physicochemical Properties of Polyaniline for Corrosion Inhibition of Mild and Stainless Steel"

_polymers, 2022, doi:10.3390/polym14153116_

Round 1

Reviewer 1 Report

Authors need to discuss the Rs Rct and CPE along with equivalent circuit diagram for Fig. 8b.

For the sake of competence to attract more attention and enhance the readership, authors need to refer these articles and discuss in introduction: ACS omega 6 (40), 26329-26337; Volume 2014 |Article ID 973653 | https://doi.org/10.1155/2014/973653; Crystals 11 (2), 124; Alexandria Engineering Journal 59 (6), 4449-4462; Chem. Mater. 2001, 13, 3, 1131–1136.

There are several typo and format errors. Please revise.

Please in index the XRD peaks.

Author Response

Reviewer 1:

The authors are thankful to the reviewer for giving time to review this article.

Comment # 1

Authors need to discuss the Rs, Rct and CPE along with equivalent circuit diagram for Fig. 8b.

Authors reply: As suggested the parameters have been discussed in the revised version.

Comment # 2

For the sake of competence to attract more attention and enhance the readership, authors need to refer these articles and discuss in introduction: ACS omega 6 (40), 26329-26337; Volume 2014 |Article ID 973653 | https://doi.org/10.1155/2014/973653; Crystals 11 (2), 124; Alexandria Engineering Journal 59 (6), 4449-4462; Chem. Mater. 2001, 13, 3, 1131–1136.

Authors reply: Thanks for nice suggestion. The suggested references have been inserted in the introduction part. (references 10, 22 and 23).

Comment # 3

There are several typo and format errors. Please revise.

Authors reply:  We have carefully revised the article for typo and formatting errors.

Comment # 4

 Please in index the XRD peaks.

Authors reply:  XRPD indexing is the process of determining the size, shape and symmetry of the crystallographic unit cell for a crystalline component responsible for a set of peaks in an XRPD pattern. Another referee suggested that your sample is amorphous in nature by the results you presented, so can’t find the crystallite size and he suggested to remove that part from XRD discussion. So we deleted that from our XRD discussion.

Reviewer 2 Report

See attached file

Author Response

Reviewer 2.

The authors are grateful to the reviewer for giving precious time to review this article. All the comments are valuable and important for improving the quality of the manuscript.

 General comments

 Comment: 1

 Application of PANI in corrosion protection is a long story and the literature on this subject is very broad. All the methods used by the authors as well as the interpretations are taken from the literature – therefore the level of novelty of the manuscript is rather low.

Authors reply: Yes, the application of PANI in corrosion protection is a long story and the literature on this subject is very broad.  However, in recent times PANI has emerged a very important corrosion protection material on which scientists are working to make it commercial material for steel protection. We are also working to enhance its inhibition properties. So we introduced a new plant material PDG which hasn’t been used before for corrosion protection of steel items. Plant material as corrosion inhibition material is new field on which very little literature is available. So due to PDG excellent properties we tried to incorporate it in PANI to enhance further its properties as corrosion protection material which we observed from results. So we claim the novelty on the basis of PDG composite with PANI. PDG haven’t been used before for any sort of application.

Comment: 2

The general conclusion of the manuscript is that the steel samples coated by the composite PDG-g-PANI corrode much slower in comparison to the bare ones. That is trivial – what else one can expect? Practically any coating reduces the rate of corrosion. Extensive investigations of a new type of the coating are worth to be undertaken only if the application of this coating offers some advantages. Is the coating which the authors investigated better than the coatings commonly used? Nothing on that subject may be found in the manuscript.

Authors reply: The respected referee pointed out that “Practically any coating reduces the rate of corrosion” yes but it is related to the degree of corrosion rate, some coatings reduce the rate of corrosion more while others less. On the other hand a corrosion protection coating must fulfill all the needs for an excellent coating like insulating behavior, reversible nature, mechanical strength, high thermal stability, self-healing ability and low absorbing nature. Our coating material fulfill almost all the necessary requirement and offers tremendous corrosion protection upto 99 % both in saline and acidic medium, proved from electrochemical and weight loss measurements. At last the Referee asked for comparative study so literature regarding previous study has been incorporated into introduction part line 67 to 76.

Comment 3:

The materials used by the authors are not well defined. Authors used in the synthesis of their coatings green material produced from the plum trees planted in a particular place. Such kind of a material may change its properties depending on many factors, weather for example. Is the material authors used representative for a more broad kind of materials? The same concerns diesel used in the synthesis. The composition of the diesel may change depending on the producer and fuels usually contain additives, which may influence the synthesis. 

Authors reply: The specification of Prunus domestica gum is defined in (Purification and physicochemical characterization of Prunus domestica exudate gum polysaccharide https://doi.org/10.1016/j.carpta.2020.100003).  FTIR, UV, XRD and SEM of our gum confirmed the results as reported in the concerned paper. To point out the location of the plant trees from which gum was obtained was necessary from scientific point of view. For specific specie of plant the gum chemical composition will remain same whatever the location of plant may be, however the amount of gum exudates may change depending on environmental conditions, but still the main components will remain the same. On the other hand the there is no clear effect of diesel on the physicochemical properties of final product as it was observed in our previous work, because commercial diesel obtained from different pumps almost contain same components of  hydrocarbons with negligible difference. In our previous work we haven’t observed interaction of solvent diesel on the structure of final product which was polyaniline (Materials 2019, 12, 1527; doi:10.3390/ma12091527).

Comment 4:

Manuscript is written very lengthily. All measured data are practically presented in triplicate: in a figure, in a table and discussed in detail in the text. Some parts of the text should be shortened significantly.  For example the part describing the FTIR spectra. The only conclusion from this part is, that composite is composed of gum and PANI. Absorbance is an additive property and the spectrum of a composition should show all peaks of the components. It was really observed. Table 1 and the part of the text describing the assignment of the peaks are not necessary.

Authors reply : We are agreed with respected Referee about the length of the data. In corrosion study it is necessary to present the data in triplicate as graph, table and discussion has their own importance, as all three will clear the image about the corrosion behavior. However we accept the respected Referee suggestion about FTIR Table 1 deletion and the table has been deleted from the article.

Particular comments:

Comment 5:

Authors attached the file with supplementary information but they do not refer to it in the main body of the manuscript

Authors reply:  Correction has been done in the revised file.

Comment 6:

 Line 60. “Eddy et al. [reference]stated that gums have good corrosion………….”  What does it mean [reference]?

Authors resply: The correction has been made and the reference 18 at the end of paragraph refers to Eddy et al. statement.

Comment 7:  

Line 110: “UV-Vis 1800 Shimadzo…” It should be Shimadzu

Authors reply:  Correction has been done.

Comment 8;

 Line 126: “Reference 3000 ZRA potentiostat/galvanostat….” What does it mean

“Reference”?

Authors response: This is the name of potentiostat which was used for electrochemical studies.  It is manufactured by Gamry, USA.

Comment 9:

 Line 127: “A silver plate and saturated calomel electrode (SCE) were used as counter and reference electrodes, respectively.” Silver is absolutely unsuitable as a counter electrode, especially in H2SO4 solutions. It is not noble enough, when polarized anodically it would dissolve.  Authors should check the possible influence of silver on the course of electrochemical experiments.

Authors reply: Yes indeed silver is less inert as compared to platinum and gold electrode. As auxiliary electrode just keep the current balance in the cell. So any inert electrode like platinum, gold or silver can be used. On the other hand the potential window and scan rate we applied during our experiment is not enough to make silver anodized or make it dissolved. As for as its dissolution is concerned so we are working with silver electrode as auxiliary electrode for last 6,7 years in corrosion studies but we haven’t observed its dissolution, even we worked with silver plate in 5 M H2SO4  solution but still it act as inert.

Comment 10.

Line 128: “Before coating, steel electrodes were polished using an emery paper of different sizes and made smooth with alumina powder.” Please give the grade of the papers and the alumina powder!

Authors reply:  The grades of polishing cloth and alumina powder are given in the revised file.

Comment 11.

 Line 147: ” …………..both coated and bare electrodes were dipped in salt solution and kept for specific amount of time. The same process was also repeated in open air as well.”  Where the electrodes were kept for specific amount of time if not in open air?

Authors response: There were two studies of weight loss method. First the coated and uncoated electrodes were kept in 3.5 % salt solution for specified period of time, and from weight loss measurement the corrosion rate was determined.

In 2nd case  the same procedure was applied but in open air. Both the electrodes were dipped in 3.5 % NaCl solution for 30 minutes and then kept in open air. The electrodes were sprayed with salt solution after each three days and left in open atmosphere for 2 months. After 2 months uncoated mild steel electrode was rusted while coated electrode was remained almost the same after removing the material.

The corrosion rate for both coated and uncoated electrodes were determined by using the following equation.

Cr =

Comment 12:

 Line 182: “Interestingly, the UV-Vis spectra of PDG-g-PANI exhibits four peaks having

three characteristics peaks of pristine PANI and a respective peak of PDG.”  Why

interestingly? What else can be expected?

Authors reply: Actually it has been observed that in composite formation or doping phenomenon the characteristic peaks of parent chain only appears with blue or red shift in UV/Visible spectra. However here PDG-g-PANI composite the characteristic peaks of both components appears that’s why we use word “interestingly”.

Comment 13.

Line 227: “XRD patterns of PDG-g-PANI samples (Figure 5) exhibited a broad peak in the 2Ө region of 18-19o [31], thus indicating the amorphous nature of the composites [23, 32].Debye Scherer formula was used to calculate the crystallite size (D) as given below.” Application of the Debye Scherer formula in the case of an amorphous substance make no sense.

Authors reply: We are agreed with Referee comment and the concerned part has been deleted from the article line 240-250.

Comment 14:

Writing EOCP make no sense. Write simply OCP. “P” in “OCP” means potential, so in EOCP potential appears twice.

Authors reply: We appreciate referee suggestion and the correction has been made in the whole article line 282, 284.

Comment 16:

 Line 270: “Generally, the occurrence of a high value EOCP is correlated with the presence of a higher corrosion protection layer.” What the authors mean “higher corrosion protection layer”?

Authors reply: This means “a layer with higher corrosion protection ability”.  Correction has been made in the revised file.

Comment 17:

Line 269: “A prominent positive shift in the EOCP of PDG-g-PANI coated MS indicated that the surface of MS is passivated under the anodic protection effect [33].” Authors should check what is the passivation potential of MS, tracing a cyclic voltammetry curve for the uncoated MS. It seems to me that the observed potentials are still in the range of active dissolution, not in the range of passivation.

Authors reply: the authors agree with this comment that passivation potential haven’t achieved in OCP data for that a long term experiment will be performed  and will be reported in future.

Comment 18:

 Line 292: “PDG-g-PANI coated electrode showed higher charge transfer resistance as compared to the uncoated sample, indicating that the surface of MS is well protected against corrosion under the prevailing chemical environment [37].” What does the authors mean “prevailing chemical environment”? It is not clear.

Authors reply: The “prevailing chemical environment” word has been changed for clarification with “chemical corrosive environment” line 309.

Comment 19:

 In figure 8 the ordinate of the left part has the dimension Ohm cm-1, the ordinate of the right part has the dimension Ohm, whereas both should have the dimension Ohm cm2.

Authors reply: The correction has been made as suggested.

Comment 20:

In figure 9 the ordinate has the dimension A cm-1, whereas it should have the dimension Acm-2.

Authors reply: The correction has been made as suggested.

Comment 21:

Generally the spontaneously established potential of a corroding electrode (OCP) should be equal to corrosion potential. However authors observed significant differences between OCP and Ecorr for both uncoated and coated electrodes. What is the  reason? (Lines 260, 267, 303 and 315).

Authors reply: Yes the referee is correct that OCP must be equal to corrosion potential. But for that OCP measurement must be carried out till the potential becomes stable which can take days or weeks which were quite impossible due unavoidable conditions in our lab such as two to three times power outages on daily basis. So unfortunately we carried out the OCP measurement for a specified period of time in order to confirm the behavior of coating only. The difference of potential between uncoated and coated steel electrode conclude the inhibition property of coating.

Comment 22:

Line 355: “The data reflect amazing corrosion protection behavior of PDG-g-PANI on both MS and SS.” I do not think it is amazing. In the literature one may find much better protective coatings.

Authors reply: We got 97 and 99 percent inhibition efficiency for both MS and SS which are considerably more as compared to that reported in literature. Similarly our composite is an ecofriendly due to insertion of green material in PANI. That’s why called it amazing results.

Comment 23:

Line 364: “The results obtained are given in Figure 11, which showed that corrosion rate was increased with rise in temperature.” “Was increased” is a grammatical error. It should be either “increased” or “have increased.

Authors reply: Thanks for pointing out mistake.. The correction is done in the revised file.

Comment 24:

 In many tables corrosion current has the dimension μA instead μA cm-2.

Authors reply: The correction is done in the revised file..

Comment 25:

Line 362: 50 Co should be 500 C.

Authors reply: correction .is done in the revised file.

Comment 26:

 Line 379: “Ea value in the presence of PDG-g-PANI coated MS is higher than that of uncoated MS in salt solution, which can be attributed to the decrease in surface area available for corrosion.” Not true! Ea does not depend on the available area.

Authors reply: Actually we have followed a published paper for this statement which are given as  (A.Biswas et al./Applied Surface Science 353 (2015) 173–183) (https://doi.org/10.1016/j.apsusc.2015.06.128) page number 178.

Comment 27:

 Line 387: “A concentrated H2SO4 solution having concentration of 1 M was used.” The

concentration of 1 M is not so high to call it “concentrated”.

Authors reply: We accept the referee suggestion and concentrated word has been deleted. line 401,402.

Comment 28:

Line 394: “After coating composite on MS both the anodic slope and cathodic slope

shifted to negative side with 0.077 and 0.200 V/decade.” This sentence is

incomprehensible. They simply decreased, not shifted.

Authors reply:  Correction has been done in the revised file.

Comment 29:

Figure 11c is not an Arrhenius plot!

Authors resply: Thanks for pointing out our mistake. Correction is done in the revised file.

Comment 30:

 Table 7. “Corrosion kinetic parameters of uncoated and coated MS before and after immersion.” I do not see any “corrosion kinetic parameters” in this table.

Authors reply:  Thanks for pointing out our mistake. Correction is done in the revised file.

Comment 31:

Application of gravimetric measurements in the case of coated samples is unjustifiable, because coating always adsorbs electrolyte which distort the results.

Authors reply: The weight of electrode was taken before coating and again after coating was removed in gravimetric analysis. We haven’t taken the weight of coated electrode. First the uncoated electrode weight was taken. Then it was coated with PDG-g-PANI and was dipped in NaCl solution. After 14 days the coating was removed and weight of electrode was taken again.

Round 2

Reviewer 2 Report

See attached file.

Round 3

Reviewer 2 Report

Figure 9. Ordinate should be A cm^-2.